# Developing a Deep Learning Model to Evaluate Bulbar Conjunctival Injection with Color Anterior Segment Photographs

**DOI:** 10.3390/jcm12020715

**Published:** 2023-01-16

**Authors:** Shanshan Wei, Yuexin Wang, Faqiang Shi, Siman Sun, Xuemin Li

**Affiliations:** 1Beijing Keynote Laboratory of Ophthalmology and Visual Science, Beijing Institute of Ophthalmology, Beijing Tongren Eye Center, Beijing Tongren Hospital, Capital Medical University, Beijing 100069, China; 2Beijing Key Laboratory of Restoration of Damaged Ocular Nerve, Department of Ophthalmology, Peking University Third Hospital, Beijing 100191, China; 3State Key Laboratory of Virtual Reality Technology and Systems, School of Computer Science and Engineering, Beihang University, Beijing 100191, China; 4Peking University Health Science Center, Peking University, Beijing 100191, China

**Keywords:** deep learning, bulbar conjunctival injection, artificial intelligence, automated approach

## Abstract

The present research aims to evaluate the feasibility of a deep-learning model in identifying bulbar conjunctival injection grading. Methods: We collected 1401 color anterior segment photographs demonstrating the cornea and bulbar conjunctival. The ground truth was bulbar conjunctival injection scores labeled by human ophthalmologists. Two convolutional neural network-based models were constructed and trained. Accuracy, precision, recall, F1-score, Kappa, and the area under the curve (AUC) were calculated to evaluate the efficiency of the deep learning models. The micro-average and macro-average AUC values for model grading bulbar conjunctival injection were 0.98 and 0.98, respectively. The deep learning model achieved a high accuracy of 87.12%, a precision of 87.13%, a recall of 87.12%, an F1-score of 87.07%, and Cohen’s Kappa of 0.8153. The deep learning model demonstrated excellent performance in evaluating the severity of bulbar conjunctival injection, and it has the potential to help evaluate ocular surface diseases and determine disease progression and recovery.

## 1. Introduction

The bulbar conjunctival injection is a common clinical indicator of ocular diseases. It appears in various forms of ocular irritation, infection, and inflammation [1]. The grading of bulbar conjunctival injection is tightly associated with the diagnosis, severity assessment, and rehabilitation of ocular diseases. The grading of bulbar conjunctival injection is commonly based on the degree of redness, the area involved, and whether or not it is focal or diffused [2]. Various standardized grading scales have been applied to evaluate the severity of bulbar conjunctival injection, including the McMonnies/Chapman-Davies scale, the Cornea and Contact Lens Research Unit grading scale, and the Institute for Eye Research scale and the validated bulbar conjunctival injection scales [3,4,5,6]. These grading systems depend on subjective judgment, the main limitation of which is the inter-grader and intra-grader variability that reduces evaluation accuracy.

Considering these limitations, objective bulbar conjunctival injection assessment tools have been developed to solve problems [7,8]. Park et al., used contrast-limited adaptive histogram equalization to enhance the blood vessel and segmented the vessel with a certain threshold for scoring [9]. Most studies utilized digital edge detection and color extraction [10,11,12,13]. Wolffsohn’s studies determined the amount of red-channel activity relative to total channel activity to grade ocular redness [10,14]. Amparo’s teams applied a more comprehensive algorithm to read the RGB value on pre-treated conjunctiva photographs digitally, convert them to HSV space, and calculate the redness score [15]. These methods require image pre-processing, and the result might be affected by background color and other redness other than conjunctive injection.

With the rapid development of deep learning, a convolutional neural network-based artificial intelligence has been quickly introduced into the field of ophthalmological image analysis, including diabetic retinopathy [16], age-related macular degeneration [17], glaucoma [18], and dry eye [19,20]. A convolutional neural network could conduct classification directly from photographs by training the algorithm with labeled images. The neural network could extract features layer by layer automatically rather than recognize prespecified features. Currently, few studies have focused on conjunctival injection classification with the convolutional neural network. The primary aim of the present research was to evaluate the feasibility and accuracy of a convolutional neural network-based deep learning model in grading bulbar conjunctival injection.

## 2. Materials and Methods

### 2.1. Subjects and Dataset

This study was conducted following the Declaration of Helsinki, and the protocol was approved by the institutional review board of the Peking University Third Hospital. The individual information for each image was removed for privacy, and the image could not be linked to an individual patient.

We retrospectively collected 1401 color anterior segment photographs from 1179 patients between March 2019 and 2020 in the Department of Ophthalmology of Peking University Third Hospital. All of the photographs were captured in a standardized method by the same examiner using a digital slit lamp camera system consisting of a BX-900 Eyecap system (Haag-Streit, Koeniz, Switzerland) and a Canon EOS 40D (Canon Inc., Tokyo, Japan). Simultaneously the upper and lower eyelid was slightly opened manually to expose the conjunctiva better. We applied 10× magnification and focused on the conjunctiva with diffused light during photography. The photograph should contain the entire temporal and nasal conjunctiva and the cornea in the middle of the image. The color anterior segment photographs were taken in the clinic to assess different ocular surface diseases, including dry eye, conjunctivitis, keratitis, etc. We excluded images from patients with severe ocular trauma, ocular surface surgeries, and diseases influencing the observation of conjunctiva, including severe ptosis and large conjunctiva tumor or unclear images.

The ground truth of bulbar conjunctival injection grading was established by three independent ophthalmologists with more than five years of working experience. Initially, three graders independently graded the bulbar conjunctival injection using the Cornea and Contact Lens Research Unit scale from grade 0 to 4, as shown in Figure 1. The Cornea and Contact Lens Research Unit scale is a 5-point bulbar conjunctival injection photographic scale: 0, white eye; 1, very slight; 2, slight; 3, moderate; 4, severe [21]. The order of the photographs was disrupted for two graders. After summarizing the results, the discrepancies in the image screening and labeling were resolved by the fourth expert ophthalmologist with more than 20 years of clinical experience.

### 2.2. Model

To grade bulbar conjunctival injection (BCI), we applied ResNet34 for BCI Model for bulbar conjunctival injection grading from 0 to 4 for each sample. With its novel residual connection design, the ResNet model can train a deeper convolutional neural network without worrying about vanishing and exploding gradients [22]. Therefore, a deeper convolutional neural network model, ResNet34, was leveraged in our study for a more complicated five-grade classification for the severity of bulbar conjunctival injection. With more convolutional and batch normalization operations, ResNet34 can identify and locate more complicated data features to recognize and solve intricate image problems.

We randomly allocated the images into three non-overlapping sub-datasets, a training dataset (60%), a validation dataset (20%), and a testing dataset (20%), to develop and evaluate the deep learning algorithm. We carried out random sampling individually to avoid data from the same patient being allocated into the same dataset, which would cause biased estimates in the deep learning model performance. The image in the training dataset was applied to train the deep learning model, and the validation dataset was aimed to tune the hyperparameter. The testing dataset evaluated the performance of the model in grading bulbar conjunctival injection.

To train the model, the entire dataset was normalized according to the mean and standard deviation of the pixel values in the images in the ImageNet training dataset. Then, the original image with 3456 by 2304 pixels obtained from the digital slit lamp camera system was padded with blank pixels and resized into a square image of 512 by 512 pixels. We also leveraged data augmentation methods to enhance and expand the training dataset. A lateral inversion was first applied to all training images to increase the size of the training dataset. Next, any one color jittering, random lighting via subtle changes in image brightness and contrast of up to 10%, scaling using a randomly selected multiplier in the range [0.8, 1.2] and cropping, rotating using a randomly selected angle in the range [−15°, 15°], and shifting was applied with 50% probability when the image was forwarded to the model training process. For training datasets with severe category imbalance, most importantly, we used the method of category balance re-sampling to train our models to prevent the model from favoring only specific categories, recognizing only labels with large numbers of samples in the training dataset, and ignoring labels with small numbers of samples.

Our study leveraged the same parameter settings for the two models described above. All pre-trained parameters from ResNet were unfrozen and retrained, and different learning rates from 0.0001 to 0.01 were used in different parts of the model. We used larger learning rates for the later layers to learn high-level semantic features in the dataset, while a lower learning rate was beneficial for maintaining the underlying feature extraction capabilities of the model transferred from the large-scale ImageNet dataset. A mini-batch size of 8 was used for trading off training efficiency and the limitations of the machine hardware. The cross-entropy loss function and Adam optimizer provided by the PyTorch framework were also employed with the default best parameters to train our models. For better training efficiency, an early stop mechanism was used with a setting of 100 epochs.

Our deep learning models were implemented with Python 3.8 and the PyTorch 1.3 DL framework. The training and evaluation platform was configured with two Intel Xeon E5-2650 CPUs, four Nvidia Tesla V100 GPUs, and 128 GB of memory.

### 2.3. Evaluation

To evaluate the performance of our deep learning algorithm for grading bulbar conjunctival injection, we quantitatively compared the results predicted by the deep learning algorithm with those annotated by experienced ophthalmologists on the validation dataset.

Accuracy, precision, recall, F1-score, and Cohen’s kappa were calculated:$$\mathrm{Accuracy}=\frac{TP+TN}{TP+FP+TN+FN}$$
$$\mathrm{Precision}=\frac{TP}{TP+FP}$$
$$\mathrm{Recall}=\frac{TP}{TP+FN}$$
$$F1\text{-}\mathrm{score}=\frac{2TP}{2TP+FP+FN}$$
$${\mathrm{Cohen}}^{\prime }s\;\kappa \mathrm{appa}=\frac{p_{0}-p_{e}}{1-p_{e}}$$
$$p_{e}=\frac{a_{1}*b_{1}+a_{2}*b_{2}+\ldots +a_{i}*b_{i}}{n*n}$$
where *a_i_* denotes the total number of labeled class *i* and *b_i_* is the total number of predicted class *i*. *P*_0_ is the sum of the number of correctly classified samples in each class divided by the total number of samples. True positive, *TP*; false positive, *FP*; true negative, *TN*; false negative, *FN*.

The areas under the receiver operating characteristic curves were estimated to evaluate the performance of models. In particular, the micro-receiver operating characteristic curves and the areas under these curves were calculated globally by considering each example of the label indicator matrix as a label, while the macro-receiver operating characteristic curves and the areas under these curves were calculated for each label after its unweighted mean was determined. Therefore, the macro metrics did not take label imbalance into account. We generated confusion matrices to show the results of the DL model and manual labeling.

## 3. Results

There are 49 patients with a score of 0, 561 patients with a score of 1, 466 patients with a score of 2, 200 patients with a score of 3, and 125 patients with a score of 4. The ROC curves for the deep learning model grading bulbar conjunctival injection are shown in Figure 2a. The micro-average and macro-average AUC were 0.98 and 0.98, respectively. The AUC value for grading bulbar conjunctival injection as 0 to 4 were 0.98, 0.98, 0.98, 0.97, and 0.99. The confusion matrices comparing the performance between the bulbar conjunctival injection grading model and the ground truth is shown in Figure 2b. The confusion matrices summarized the count of each grade in the bulbar conjunctival injection identified by the model or ophthalmologists. The agreement between the predicted label and ground truth was high among all grades in conjunctival injection judgment. The accuracy, precision, recall, F1-score, and Cohen’s kappa of the model grading bulbar conjunctival injection and the performance of human ophthalmologists are demonstrated in Table 1. The bulbar conjunctival injection grading model demonstrated high accuracy of 87.12%, with a precision of 87.13%, a recall of 87.12%, an F1-score of 87.07%, and Cohen’s kappa of 0.8153. The performance metrics were superior to two ophthalmologists.

## 4. Discussion

Our study established deep learning models to grade bulbar conjunctival injection through convolutional neural networks. The results demonstrated that the established model performed better than some ophthalmologists. The present research shows the possibility of applying DL models for bulbar conjunctival injection grading with color anterior segment photography. The bulbar conjunctival injection is important in diagnosing and evaluating ocular inflammatory and infectious diseases. Thus, it is essential to evaluate the severity of bulbar conjunctival injection objectively. The grading scores applied in clinical practice generally range from zero to four or five [3,5,16]. However, some images might be better regarded with scores between two grades. Furthermore, the subjective grading in the clinic depends on the experience and individual perceptions of each doctor, and intra- and inter-grader inconsistencies might cause inaccuracies in grading. Hence, it was necessary to develop an objective and continuous quantitative evaluation system.

Several studies have reported various image analysis techniques for grading bulbar injection. Downie et al. [16] reported a method that measured bulbar conjunctival injection objectively by calculating the percentage of pixels in the regions of interest of the images. Some studies developed bulbar conjunctival injection image analysis software to extract blood vessels based on thresholding [10,23,24], the percentage of red in the image [25,26], image smoothing, and edge detection [10,27]. However, these methods require image pre-processing, and photographs with poor sharpness and unsuitable background color might not be appropriately identified with this technique. Recently, the convolutional neural network-based deep learning model has been widely applied in medical image recognition. The neural network could effectively extract the features in the images automatically to complete various tasks, including classification, grading, and lesion recognition [16,17,18,19,20]. By training the model based on a convolutional neural network with certain labeled images, the models can automatically identify similar labels in the images effectively.

The developed model provides an accurate and automatic grading of the severity of bulbar conjunctival injection. The model could be potentially applied in several clinical and experimental scenarios to assess bulbar conjunctival with repeatability and reliability. Firstly, the researcher could apply the model to enroll patients with a certain level of bulbar conjunctival injection rapidly in clinical trials investigating the treatment targeting dry eye, conjunctivitis, keratitis, and uveitis. Secondly, the model facilitates a rapid evaluation of ocular surface disease rehabilitation during the follow-up period based on bulbar conjunctival injection recovery. Thirdly, the model could be applied in community-based screening and non-professional health examinations to indicate referral. With further improvement, the model might be embedded in mobile phone applications so that the bulbar conjunctival evaluation could be performed with selfies anywhere and anytime in the future.

Certain limitations exist in the present research. First, the proportions of image bulbar conjunctiva injection grades 0 and 4 were relatively low. This might cause bias in the model identifying certain bulbar conjunctiva injection grades. Second, the model was validated with images similar to the training dataset for photograph brightness and color. Further external testing with images from other machines and centers should be conducted to verify the generalization capability of the model. Third, the actual injection may not be accurately represented in the photograph. The performance of the model may be inferior compared with slit lamp microscope observance. The present model could only identify bulbar conjunctival injection grading based on anterior segment photography, and the identification of these two signs could not be directed to certain ocular diseases. Further evidence and clinical manifestations are required to support the diagnosis of the disease. In future investigations, we would further validate the model with an external dataset and test the model in various ocular diseases that could improve the facility of the model.

In conclusion, this convolutional neural network-based deep learning model demonstrated robust performance in evaluating the severity of bulbar conjunctival injection. The performance was similar to or possibly better than some human ophthalmologists. The present research provides the potentiality of a deep learning model to facilitate the assessment of bulbar conjunctiva in diagnosing ocular surface diseases and determining disease severity and prognosis.

## Figures and Tables

**Figure 1 jcm-12-00715-f001:**
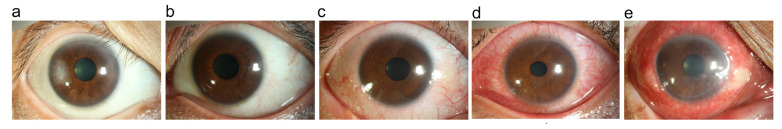
Grade of bulbar conjunctival injection severity based on the Cornea and Contact Lens Research Unit scal. (**a**): Grade 0, (**b**): Grade 1, (**c**): Grade 2, (**d**): Grade 3, (**e**): Grade 4.

**Figure 2 jcm-12-00715-f002:**
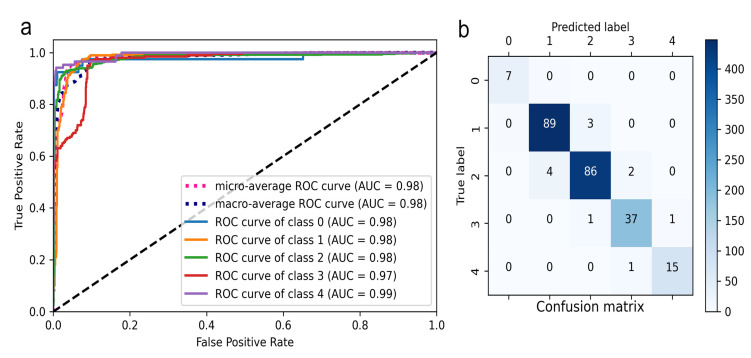
Performance of the deep learning model grading bulbar conjunctival injection. (**a**). Receiver operating characteristic curves for bulbar conjunctival injection grading (true positive rate-*y* axis; false positive rate-*x* axis); (**b**). Confusion matrix for the deep-learning model and manual labelling. The confusion matrices summarized the count of each grade in the bulbar conjunctival injection identified by the model or ophthalmologists. The agreement between the predicted label and ground truth was high among all grades in conjunctival injection judgment.

**Table 1 jcm-12-00715-t001:** Performance of the deep learning model grading bulbar conjunctival injection.

	Accuracy	Precision	Recall	F1-Score	Cohen’s Kappa
Ophthalmology 1	0.7202	0.7671	0.7202	0.7194	0.6043
Ophthalmology 2	0.8728	0.8904	0.8728	0.8775	0.8217
Ophthalmology 3	0.7679	0.8079	0.7679	0.7695	0.6697
Model	0.8712	0.8713	0.8712	0.8707	0.8153

## Data Availability

The analysis data used in this study are available from the corresponding author upon request.

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
