# Peer review of "Developing a Deep Learning Model to Evaluate Bulbar Conjunctival Injection with Color Anterior Segment Photographs"

_jcm, 2023, doi:10.3390/jcm12020715_

Round 1

Reviewer 1 Report

The work “ Developing a deep learning model to evaluate subconjunctival hemorrhage and bulbar conjunctival injection with color anterior segment photographs” is a deep learning technology work using CNN for developing a subconjunctival hemorrhage and conjunctival injection detection and grading algorithm with sophisticated methodologies, including classifications with many individuals, cropping, image augmentation, randomization, cross-validation, calculating interobserver agreement as well with impressive scores and should be published after edits.
The analysis of an ordinal variable such as the conjunctival injection grading is different from that of a binary classification, which is basically nominal, between subconjunctival hemorrhage presence or absence. Both classifications are of importance for ophthalmology and potentially telemedicine purposes. Moreover, the bulbar hyperemia grading is far more elaborate and having both methods presented in the paper may dilute each other relevance. I suggest considering leaving only bulbar conjunctival injection on this paper, and on another paper including subconjunctival hemorrhage.

The authors may choose to add a comparison of patients with high-grade conjunctival injection (perhaps 4 and 5) with subconjunctival hemorrhage pictures to detect the capacity of the algorithm for detection and exclusion of hemorrhage with high-grade bulbar hyperemia. This can also be left for the external validation that the authors comment they will do in the future. This is especially useful for telemedicine purposes.

The use of English can be enhanced and there are many editing mistakes such as lack of spaces (“ “) between words, and phrases highlighted in red (probably as tracked changes) which should have been corrected prior to submission. The manuscript needs a substantial work including a full grammar review. I would suggest the authors revise the language and the writing fluency throughout the manuscript to improve its clarity, style and grammar

The reader would benefit from observing pictures used from the patients from the study used as ground truth for both subconjunctival hemorrhage and bulbar hyperemia.

Please elaborate on the cropping method used. Sometimes cropping everything but the region of interest (i.e: cropping the cornea, and eyelids for example, leaving only the conjunctiva increases scores)

Were other backbone CNN models tested other than resnet 18 & 34? Analyzing others would give more strength to the study.

The Cohen kappa data was presented for the totality/average of the bulbar conjunctival injection grade? Was it different for any specific grade? In other words, was the machine better at detecting a certain grade of hyperemia than other grades, in comparison to humans?

Line 319: No data supports that these algorithms predict prognosis.

Reviewer 2 Report

I congratulate all the authors who have been engaged in such an invention. However, few statements must be added to clarify the reasoning behind the study.

1.     How does the current algorithm benefit from the available techniques?

2.     What is the possible cost-benefit ratio in such a technique?

3.     The language needs polishing.

4.     Please arrange the references in a single fashion.
